

# Soil fertility evaluation and spatial distribution of grasslands in Qilian Mountains Nature Reserve of eastern Qinghai-Tibetan Plateau

Qiang Li, Junyin Yang, Wenhao Guan, Zhigang Liu, Guoxing He, Degang Zhang and Xiaoni Liu

College of Grassland Science, Gansu Agricultural University/Key Laboratory of Grassland Ecosystem of the Ministry of Education, Lanzhou, China

## ABSTRACT

The study assessed the overall soil characteristics of grasslands on Qilian Mountains and rated the soil nutrient status with classification standard of the second national soil survey of China. Nemerow index method was used to evaluate the soil fertility of different grassland types. GIS was used to analyze the spatial distribution of the soil nutrients and provided a database for the grassland's ecological protection and restoration. The study graded the soil organic matter (SOM), total N, and available K at level 2 (high) or above for most regions, available soil-P at level 4, while the soil bulk density, total porosity and pH were $0.77$–$1.32$ g cm$^{-3}$, $35.36$–$58.83\%$ and $7.63$–$8.54$, respectively. The rank of comprehensive soil fertility index was temperate steppe (TS) > alpine meadow (AM) > alpine steppe (AS) > upland meadow (UM) > alpine desert (AD) > lowland meadow (LM) > temperate desert steppe (TDS) > temperate desert (TD). The areas with high, medium and low soil fertility accounted for $63.19\%$, $34.24\%$ and $2.57\%$ of the total grassland area. Soil fertility of different grassland types had different main limiting factors, for instance, the pH, total N and SOM were the main factors limiting soil fertility in LM, while pH and available P were the main factors limiting soil fertility in UM, AM, TS and AS. In summary, the grassland soil fertility was generally at the mid-upper level, and the main limiting factors were found in the different types of the grasslands and their spatial distributions were figured out. Our findings also indicated that the typical grasslands and meadows may require phosphorus application, while for desert grasslands, both nitrogen and phosphorus were required to improve their comprehensive soil fertility and grassland productivity.

## INTRODUCTION

Qilian Mountains Natural Reserve is one of the most sensitive regions under global warming and an important ecological security barrier in northwestern China (*Wang, Ren , & Zhang, (2001)*. Grassland ecosystem is the largest ecological system in Qilian Mountains Natural Reserve, which accounts for 74.3% of the total area and plays an critical role in maintaining

Corresponding author
Xiaoni Liu, liuxn@gsau.edu.cn

biodiversity, water conservation and ecological balance of the natural reserve (*Li et al., 2019*). In the last decades, the grassland ecosystem have been severely damaged because of climate change, human activities and mismanagement in this area. Understanding the current status of grassland soil in Qilian Mountains is of great significance to the health and sustainable maintenance of grassland ecosystem. Due to differences in topography, precipitation and temperature, the distribution of same grassland type is very patchy, discontinuous and irregular. Previous studies found that different grassland types have large differences in soil nutrients due to the differences in vegetation types and utilization methods (grazing, water conservation and sand fixation) (*Fayiah et al., 2019*; *Chen et al., 2019*).

Soil fertility directly affects the health of grasslands and is also influenced by grassland vegetation (*Hao et al., 2020*). Without human disturbance, the growth and distribution of grassland vegetation is strongly affected by soil fertility apart from climate (*Wang et al., 2016*; *Harpole, Potts & Suding, 2007*). Soil fertility affects not only the growth of grassland vegetation, but also the grassland ecosystem health (*Ma et al., 2019*). Therefore, better understanding and proper evaluation of soil fertility characteristics are of great significance to restoration of degraded vegetation, and improvement of fragile grassland ecosystem (*Su et al., 2019*; *Jin et al., 2018*).

Grassland soil fertility plays a key role in supporting grassland ecosystem services (*Clanet, 1980*; *Hu et al., 2018*; *Qu et al., 2016*). Soil organic matter, available nitrogen, available phosphorus, available potassium, soil bulk density and pH are the important components of soil fertility, while their contents directly affect grassland vegetation productivity (*Wuest, 2015*; *Li et al , 2014*). Many methods have been used for soil fertility evaluation, including Nemerow method (*Hua et al., 2018*; *Shahab et al., 2013*), analytic hierarchy process (AHP) (*Sousa et al., 2012*), subordinate function value method, etc. in which the Nemerow index is well recognized and commonly used. This method was developed by N. L. Nemerow and originally was used to evaluate water quality for pollutants (*Nemerow, 1974*). It was modified and improved by Chinese scholars (*Hua et al., 2018*; *Shahab et al., 2013*; *Zhou et al., 2017*; *Zhou et al., 2018*). The Nemerow Index is used to evaluate the comprehensive soil fertility, meanwhile the modified Nemerow Index can determine the minimum limiting factor of soil fertility. The Nemerow index method can avoid the influence of subjective factors and could highlight the influence of the worst factor of soil attribute factors on soil fertility (*Bao et al., 2012*). The Nemerow comprehensive index method also reflects the limiting factor of plant growth in ecosystem, which can improve the confidence level of the evaluation results (*An et al., 2015*; *Zhou et al., 2017*). Comprehensive evaluation of soil fertility combining with geographic information system (GIS) has been widely used to assess spatial distribution characteristics of soil nutrients, which is helpful to explore the relationship between soil nutrients and environmental factors (*Wang, Cn & Su, 2007*; *Peng et al., 2013*; *Nie et al., 2016*; *Brevik et al., 2016*; *Miller et al., 2016*).

Many studies have been carried out on the soil of degraded grassland in Qilian Mountains (*Cheng, Jia & Wang, 2019*; *Wang et al., 2018*; *Chen et al., 2016*). However; there are few studies on grassland soil fertility and its spatial distribution characteristics. Therefore, the aims of this study were to investigate the soil of different grasslands in Qilian Mountains

Natural Reserve in order to (1) analyze the distribution characteristics of soil fertility index, and (2) find out the limiting factors of grassland soil fertility to provide scientific insight for improving grassland ecological services.

## MATERIALS AND METHODS

### Study area
The study sites were located in the Qilian Mountains Nature Reserve of eastern Qinghai-Tibetan Plateau, China (94°10′−103°04′E, 35°50′−39°19′N). At horizontal direction, there are four vegetation zones in the order of forest, shrub, grassland and desert from southeast to northwest. At vertical direction, , there are three vegetation belts distributed as steppe, forest and alpine meadow from low to high altitude (3,000–5,564 m). The main types of soil are aridisols, inceptisols and entisols. The precipitation varies from 100 to 500 mm, mostly occurring from June to September. The average annual temperature is approximately −2.0 °C; the average annual relative humidity is from 20% to 70%; the annual evaporation is 1,200–1,400 mm; and the frost-free period is 90–120 days (http://www.qilianshan.com.cn).

### Sites selection and sample collection
This study sites were mainly located on the Qilian mountain natural reserve in Gansu province, China. The grassland types were temperate steppe (TS), alpine meadow (AM), alpine steppe (AS), upland meadow (UM), alpine desert (AD), temperate desert steppe (TDS), lowland meadow (LM), temperate desert (TD) (Table 1) (*NY/T 2997-2016, 2016*).

The sampling time was from July 23 to August 5, 2019, when the plants were in full bloom. The central points of the typical distribution area of the above 8 types of grasslands (AM, TS, LM, AS, UM, TDS, AD and TD) were selected as the sampling sites (Table 1). A 60-meter sample line was randomly set for each sample site and the sample spots were set for every 20-meter interval. Four soil samples were taken at each sampling site using soil drill (an auger drill) at a depth of 0–30 cm and mixed as one sample. The samples were air-dried and stored in sample bags for further test. Meanwhile, Soil bulk density was measured by a stainless steel cutting ring (5 cm diameter and five cm high) after aboveground biomass was measured, 10 cores at each site.

### Soil sample measurement methods
Soil bulk density was determined by core method (*Dong et al., 2012*). Soil total porosity was determined by water immersion weighing method (*SoilPhysicsInstitute, 1978*). Soil samples were air-dried at room temperature, and visible roots and other debris in the soil were removed. Each soil sample was sieved through a 2-mm sieve. Soil organic matter was determined by the Walkley–Black method (*Nelson & Sommers, 1996*). The measurement of total soil N was determined using a micro Kjeldahl digestion procedure (*Nelson & Sommers, 1996*). Briefly, a small amount of dried soil (passing 0.25 mm sieves) mixed with $H_2SO_4$, $CuSO_4 \cdot H_2O$ and $K_2SO_4$, heated and then made up with ammonium-free distilled water. The solution was mixed with 4 ml 40% NaOH and distilled using a Kjeldahl apparatus to release $NH_3$ for the determination of N content. Available P was extracted with sodium

**Table 1** Information of the sample sites, classification criterion of soil nutrients, grading criterion for various soil properties in the Nemerow grading method, criteria for determining the organic matter, total nitrogen and bulk density of grassland soils with different degradation degrees, descriptive statistics in various studied parameters of grassland soil in Qilian Mountains Nature Reserve.

| Type Grassland | Altitude m | longitude and latitude | Main plant species | Coverage % |
|---|---|---|---|---|
| Lowland meadow (LM) | 1364 | 39°40′35.02″N 99°8′45.09″E | *Phragmites australis* (Cav.) Trin. ex Steud, *Achnatherum splendens, sophora alopecuroides* L. | 48.33 |
| Upland meadow (UM) | 3114 | 37°11′36.47″N 102°43′42.73″E | *Potentilla anserina* L., *Poa annua* L., *Elymus nutans griseb., Melissilus ruthenicus* (L.) Peschkova, *Artemisia annua* L.. | 81.67 |
| Alpine meadow (AM) | 2977 | 37°10′48.66″N 102°47′13.83″E | *Polygonum viviparum* L., *Kobresia myosuroides* (Villars) Fiori, *Melissilus ruthenicus* (L.). *Peschkova, artemisia annua* Linn., *Saussurea japonica* DC. | 85.00 |
| Temperate steppe (ST) | 2817 | 37°22′13.68″N 102°40′44.93″E | *Poa annua* L., *Kobresia myosuroides* (Villars) Fiori, *Stipa capillata* Linn., *Potentilla anserina* L., *Artemisia annua* Linn. | 85.00 |
| Alpine steppe (AT) | 3735 | 39°16′32.99″N 97°42′52.57″E | *Stipa purpurea, kobresia myosuroides* (Villars) Fiori, *Poa annua* L., *Potentilla anserina* L., *Androsace umbellate* | 85.00 |
| Temperate desert Steppe (TDS) | 2139 | 38°57′57.23″N 99°47′41.95″E | *Sympegma regelii* Bunge, *Salsola collina* Pall., *Allium polyrhizum* Turcz, *Stipa capillata* Linn., *Ajania nematoloba* | 43.75 |
| Temperate Desert (TD) | 1358 | 39°29′29.11″N 99°18′45.00″E | *Nitraria tangutorum* Bobr, *Nitraria sphaerocarpa* Maxim, *Suaeda glauca* (Bunge) Bunge, *Sympegma regelii* Bunge | 31.67 |
| Alpine desert (AD) | 4290 | 39°15′34.39″N 97°45′6.70″E | *Rhodiola rosea* L., *Saussurea japonica* DC., *Kobresia myosuroides* (Villars) Fiori | 28.33 |

bicarbonate, and then determined by the molybdenum blue method (*SoilPhysicsInstitute, 1978*). Available K was extracted with ammonium acetate, and then determined by flame photometry (*SoilPhysicsInstitute, 1978*).

## Evaluation of soil fertility
### *Evaluation of individual indicators of soil fertility*

This study used the China second soil census standard (*National Earth System Science Data Center, 2005*, http://gre.geodata.cn) to rank the grassland soil organic matter, total N, available P, available K, pH, bulk density and total porosity indicators (Table 2) and to compare the differences between different grassland types (*Zhou et al., 2017*).

### *Comprehensive soil fertility evaluation*

Nemerow Index was used to conduct the comprehensive soil fertility evaluation as follows:

$$F = \sqrt{\frac{Fi^2 + Fimax^2}{2}}$$

where $F$ is the Composite pollution index, $F_i$ is the average value of each sub-pollution index, $F_{imax}$ is the minimum value of each sub-pollution index, and i is the number of the sampling point.

**Table 2 Classification criteria used for soil index.**

| Grades | SOM g kg$^{-1}$ | Total N g kg$^{-1}$ | Available P mg kg$^{-1}$ | Available K g kg$^{-1}$ | Interpretation |
|---|---|---|---|---|---|
| 1 | >40 | >2.0 | >40 | >0.20 | Very high |
| 2 | 30–40 | 1.5-2.0 | 20-40 | 0.15-0.20 | High |
| 3 | 20–30 | 1.0-1.5 | 10-20 | 0.10-0.15 | Upper |
| 4 | 10–20 | 0.75-1.0 | 5-10 | 0.05-0.10 | Mid-low |
| 5 | 6–10 | 0.5-0.75 | 3-5 | 0.03-0.05 | Low |
| 6 | <6 | <0.5 | <3 | <0.03 | Very low |

Meanwhile, the improved Nemerow Index was used to determine the minimum limiting factor of soil fertility and it is as follows:

$$F = \sqrt{\frac{Fi^2 + Fimin^2}{2} \cdot \left(\frac{n-1}{n}\right)}$$

where $F$ is the soil comprehensive fertility index, $F_i$ is the average value of each sub-fertility index (at one sampling point), $F_{imin}$ is the minimum value of each sub-fertility index (at one sampling point), and n is the number of participating indicators.

To improve the Nemerow comprehensive index, the minimum value of $F_i$ is used to replace the maximum value of $F_i$ in the original Nemerow comprehensive index, which highlights the impact of the soil lowest attribute on soil fertility and can reflect the minimum factor law of plant growth. In addition, the addition of the correction item $(\frac{n-1}{n})$ improves the credibility of the evaluation, that is, the more soil sub-fertility index in the evaluation, the greater the value of $(\frac{n-1}{n})$ and the higher of credibility. Meanwhile, correction item $(\frac{n-1}{n})$ also reflects the difference in evaluation results when the evaluation indicators are not equal.

According to the grading standards of soil properties in China (Table 3), the selected index parameters were standardized to eliminate numerical size differences between selected index parameters. The standardized treatment methods are as follows:

When the attribute value belongs to the level low, $c_i \leq x_a, Fi = c_i/x_a$      $(F_i \leq 1)$      (1)

When the attribute value belongs to the level upper,

$x_a < c_i \leq x_c, F_i = 1 + (c_i - x_a)/(x_c - x_a)$      $(1Fi \leq 2)$      (2)

When the attribute value belongs to the level high,

$x_c < c_i \leq x_p, Fi = 2 + (c_i - x_c)/(x_p - x_c)$      $(2 < Fi \leq 3)$      (3)

When the attribute value belongs to the level very high, $c_i > x$,      $Fi = 3$.      (4)

In above formulas, $F_i$ is the attribute division coefficient, $c_i$ is the measured value of the attribute, and $x_a$, $x_c$, and $x_p$ are the classification indexes.

The improved Nemerow index method was then used to comprehensively evaluate the grassland soil fertility in Qilian mountain.

**Table 3  Grading criterion for various soil properties in the Nemerow grading method.**

| Soil properties | | Soil bulk density g cm⁻³ | Total porosity % | pH | SOM g kg⁻¹ | Total N g kg⁻¹ | Available P mg kg⁻¹ | Available K g kg⁻¹ |
|---|---|---|---|---|---|---|---|---|
| Classification index of Nemerow | $x_p$ | 0.95 | 0.50 | 7 | 30 | 2.00 | 20 | 0.20 |
| | $x_c$ | 1.10 | 0.40 | 8 | 20 | 1.50 | 10 | 0.10 |
| | $x_a$ | 1.25 | 0.30 | 9 | 10 | 0.75 | 5 | 0.05 |

**Notes.**

$x_a$, $x_c$ and $x_p$ are the classification indexes.

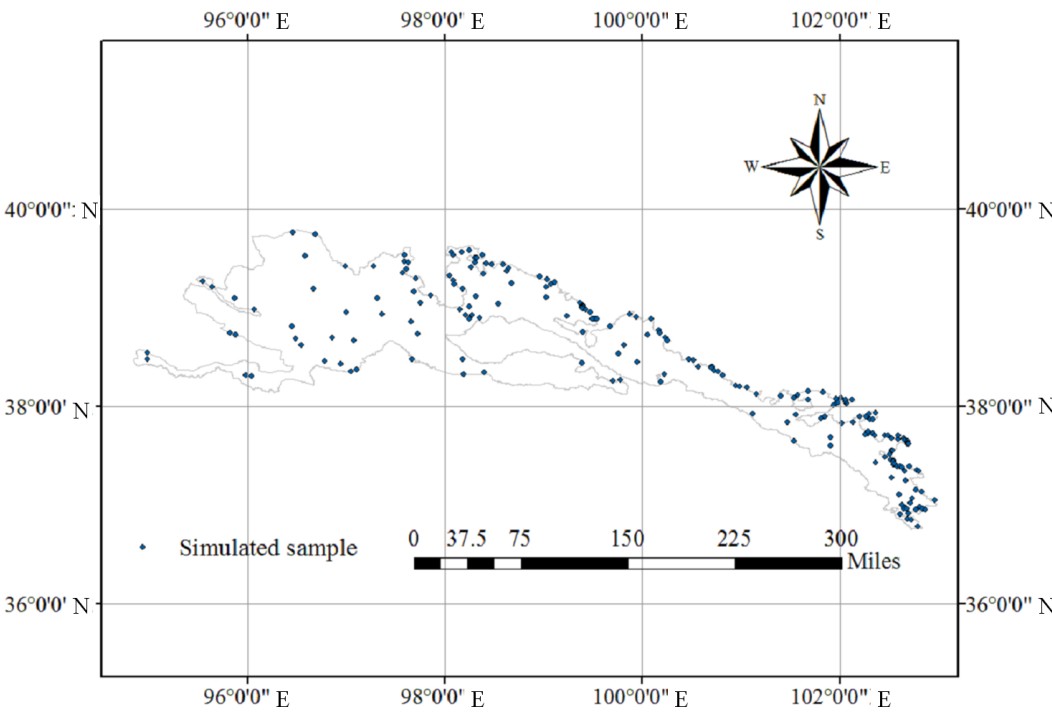

**Figure 1  The simulated samples of different grassland type patches spatial distribution.**

## Soil comprehensive fertility index spatial distribution

Analysis method based on multiple regression and residues (AMMRR) had been widely used in many studies for grassland spatial interpolation (*Liu et al., 2012*; *Guo, Ren & Liu, 2011*). This method is more accurate than many other interpolating methods and can also effectively avoids systematic errors (*Liu et al., 2012*; *Guo, Ren & Liu, 2011*). In this paper, based on the comprehensive fertility index determined by the improved Nemerow index method, the ArcGIS10.2.2 (*Nistor Mărgărit & M, 2016*) used to conduct the spatial analyses including extracting the center points of different grassland types (Fig. 1), assigning values for grassland types, performing AMMRR interpolation, and drawing the Qilian mountain grassland soil fertility index spatial distribution. The comprehensive fertility index was divided into low (<1.50), medium (1.50–2.00), and high (>2.00) (*Zhou et al., 2017*; *Zhou et al., 2018*).

**Table 4  Descriptive statistics of grassland soils in Qilian Mountains Nature Reserve.**

| Item | MIN | MAX | Mean | SD | CV% |
|---|---|---|---|---|---|
| Soil bulk density g cm$^{-3}$ | 0.77 | 1.32 | 1.01 | 0.18 | 17.88 |
| Total porosity % | 35.36 | 58.83 | 48.25 | 7.90 | 16.38 |
| pH | 7.63 | 8.54 | 8.07 | 0.38 | 4.71 |
| Total N g kg$^{-1}$ | 0.63 | 4.97 | 2.38 | 1.8 | 75.49 |
| Available P mg kg$^{-1}$ | 6.79 | 24.27 | 12.81 | 5.52 | 43.09 |
| Available K g kg$^{-1}$ | 0.21 | 1.06 | 0.40 | 0.27 | 68.00 |
| SOM g kg$^{-1}$ | 4.99 | 131.52 | 51.23 | 48.83 | 95.32 |

## Statistical analyses

Statistical analyses were conducted using SPSS (version 19.0 SPSS Inc., Chicago, IL, USA). All results were presented as mean and standard deviations. One-way ANOVA and least significant difference (LSD) tests were declared at $P < 0.05$.

## RESULTS

### Characteristics of grassland soil fertility indexes

The soil bulk density, total porosity, pH, total N, available P, available K and soil organic matter were 0.77–1.32 g cm$^{-3}$, 35.36–58.83%, 7.63–8.54, 0.63–4.97 g kg$^{-1}$, 6.79–24.27 mg kg$^{-1}$, 0.21–1.06 g kg$^{-1}$ and 4.99–131.52 g kg$^{-1}$ respectively (Table 4), and the corresponding Coefficient of variation (CV) of each index was greater than 10%.

### Soil physical and chemical properties of different grasslands

The soil fertility indexes for different type grasslands are shown in Table 5, a significant difference ($P < 0.05$) was observed between different grassland types. Soil bulk density and total porosity were in a ranking order of desert type >meadow type >steppe type. The pH was in a ranking order of TD >LM >TDS >UM >AD >AS>AM >TS. Total N was in a ranking order of AM >TS >AS >UM >AD >TDS >LM >TD. The soil organic matter was in a ranking order of TS >AM >AS >UM >AD >TDS >LM >TD. The available P was in a ranking order of LM >TS >AM >AS >UM >TD >TDS >AD. The available K was in a ranking of LM >UM >AS >TD >TS >AM >AD >TDS.

### Soil physical and chemical spatial distribution

The soil physical and chemical spatial distributions were shown in Fig. 2. For most sampling areas, the soil bulk density was 0.75–0.94 g cm$^{-3}$, the total porosity was 50–60%, the pH values were 8–9, the SOM contents were 30–134 g kg$^{-1}$, the total N contents were 2.0–5.0 g kg$^{-1}$, the available P contents were 10–20 mg kg$^{-1}$, and the available K contents were 0.3–1.5 g kg$^{-1}$.

### Soil comprehensive fertility index

The soil comprehensive fertility indexes of different type grasslands ranged from 1.01 to 2.24 (Table 6). The soil comprehensive fertility index was significantly higher in AM, UM, AS and TS than AD, significantly higher in AD than LM and TDS, and significantly higher

**Table 5  Soil physical and chemical properties in different Grassland types in Qilian Mountains Nature Reserve.**

| Grassland Type | Soil bulk density g cm⁻³ | Total porosity % | pH | Total N g kg⁻¹ | Available P mg kg⁻¹ | Available K g kg⁻¹ | SOM g kg⁻¹ |
|---|---|---|---|---|---|---|---|
| LM | 1.05 ±0.09bc | 43.65 ±4.83cd | 8.51 ±0.04a | 0.64 ±0.10d | 24.27 ±3.55a | 1.06 ±0.91a | 12.67 ±1.63cd |
| UM | 0.95 ±0.07cd | 48.73 ±2.06bc | 7.97 ±0.24b | 2.01 ±0.51c | 12.34 ±2.97b | 0.45 ±0.06b | 36.87 ±16.45c |
| AM | 0.83 ±0.09de | 51.59 ±5.45b | 7.76 ±0.26d | 4.81 ±0.13a | 13.73 ±7.54ab | 0.30 ±0.15bc | 116.46 ±28.35a |
| TS | 0.77 ±0.03df | 54.98 ±1.92ab | 7.63 ±0.10e | 4.97 ±0.78a | 14.94 ±5.69ab | 0.30 ±0.06c | 131.52 ±14.33a |
| AS | 0.91 ±0.05d | 58.83 ±2.50a | 7.83 ±0.03bc | 3.36 ±0.35b | 13.65 ±6.95ab | 0.35 ±0.08bc | 65.56 ±20.49b |
| TDS | 1.14 ±0.09b | 52.53 ±1.06b | 8.50 ±0.03a | 0.77 ±0.12d | 7.96 ±0.65b | 0.21 ±0.03c | 13.18 ±1.94cd |
| TD | 1.32 ±0.06 a | 35.36 ±4.69e | 8.54 ±0.05a | 0.63 ±0.08d | 8.81 ±2.22b | 0.32 ±0.05bc | 4.99 ±0.99d |
| AD | 1.11 ±0.06b | 40.32 ±2.18de | 7.84 ±0.04bc | 1.88 ±0.07c | 6.79 ±0.97b | 0.24 ±0.04c | 28.65 ±1.90cd |

**Notes.**

Data are presented as the mean ± SD; Different small letters in the same column mean significant difference at 0.05 level.

TS, Temperate steppe; AM, Alpine meadow; AS, Alpine steppe; UM, Upland meadow; AD, Alpine desert; TDS, Temperate Desert Steppe; LM, Lowland meadow; TD, Temperate Desert.

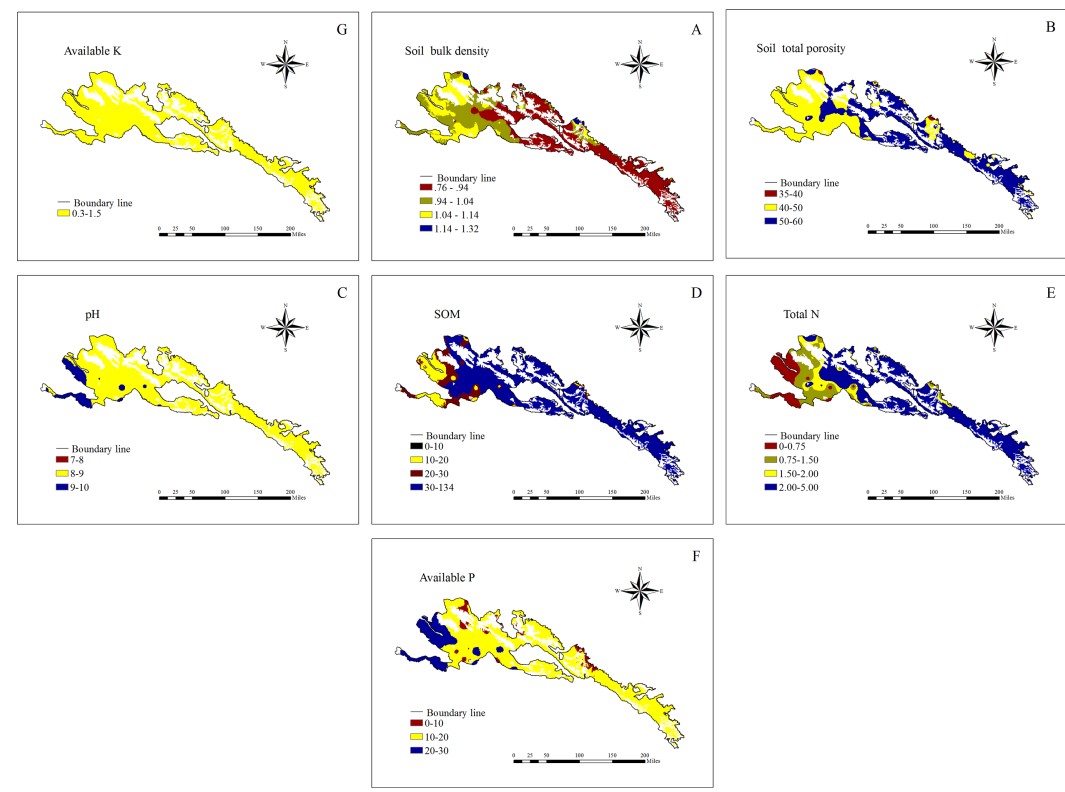

**Figure 2  Soil physical and chemical spatial distribution.**

in TDS than TD, but no significant difference was found among others. The rank of soil comprehensive fertility index was TS >AM >AS >UM >AD >LM >TDS >TD.

The soil fertility of Qilian mountain grassland was at a moderate or high level (Fig. 3). In terms of spatial distribution, the soil comprehensive fertility index was at a high level in eastern and western of Qilian Mountains, and the soil fertility in the central region was
**Table 6** Comprehensive evaluation of different Grassland Types in Qilian Mountains Nature Reserve soil fertility using Nemerow index.

| Grassland type | $F_i$ | | | | | | | $\overline{Fi}$ | F |
|---|---|---|---|---|---|---|---|---|---|
| | Soil bulk density | Total porosity | pH | Total N | Available P | Available K | SOM | | |
| LM | 2.67 | 2.37 | 1.49 | 0.85 | 3.00 | 3.00 | 1.27 | 2.09 | 1.37c |
| UM | 3.00 | 2.87 | 2.03 | 3.00 | 2.23 | 3.00 | 3.00 | 2.73 | 2.06a |
| AM | 3.00 | 3.00 | 2.24 | 3.00 | 2.37 | 3.00 | 3.00 | 2.80 | 2.17a |
| TS | 3.00 | 3.00 | 2.37 | 3.00 | 2.49 | 3.00 | 3.00 | 2.84 | 2.24a |
| AS | 3.00 | 3.00 | 2.17 | 3.00 | 2.37 | 3.00 | 3.00 | 2.79 | 2.14a |
| TDS | 1.73 | 3.00 | 1.50 | 1.03 | 1.59 | 3.00 | 1.32 | 1.88 | 1.30c |
| TD | 1.06 | 2.54 | 1.46 | 0.84 | 1.76 | 3.00 | 0.50 | 1.59 | 1.01d |
| AD | 1.93 | 2.03 | 2.16 | 2.76 | 1.36 | 3.00 | 2.87 | 2.30 | 1.62b |

**Notes.**

TS, Temperate steppe; AM, Alpine meadow; AS, Alpine steppe; UM, Upland meadow; AD, Alpine desert; TDS, Temperate Desert Steppe; LM, Lowland meadow; TD, Temperate Desert.

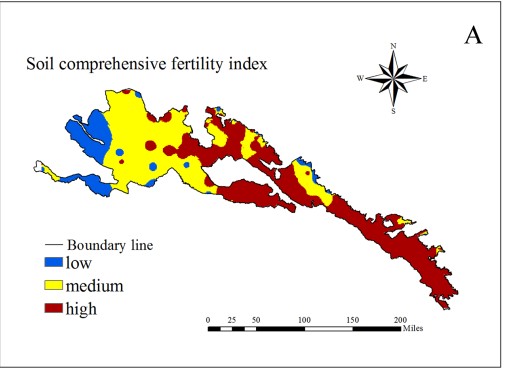 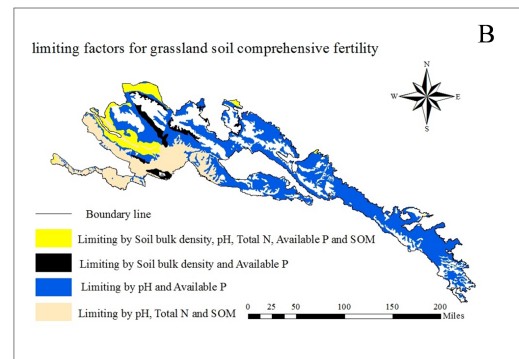

**Figure 3** Spatial distribution of (A) grassland soil comprehensive fertility index and (B) limiting factors for grassland soil comprehensive fertility.

at a moderate level. There are only a few areas where the soil fertility of the grassland was at a low level, and distributed in the marginal regions of the western and central regions. The areas with the high, medium and low soil fertility accounted for 45.60%, 41.92% and 12.46% of the total grassland area of Qilian Mountains respectively.

## Limiting factors for grassland soil comprehensive fertility

The soil fertility of different types of grasslands had different major limiting factors (Table 6). For example, the pH, total N and SOM were the main factors limiting soil fertility in LM, and pH and available P were the main factors limiting soil fertility in UM, AM, TS and AS. The Soil bulk density, pH, total N, SOM and available P were the main factors limiting soil fertility in TD and TDS. Soil bulk density and available P were the main factors limiting soil fertility in AD. The limiting factors for the comprehensive soil fertility were shown in Fig. 3.

## DISCUSS

Soil organic matter content is closely related to soil fertility and soil health. The nitrogen, phosphorus, and potassium provide essential nutrients for plant growth and development, and are the main components of soil nutrients (*Zhang et al., 2013*; *Zhou et al., 2016*). The contents of SOM and available K were graded as level 2 (high) or above according to the classification of China second soil census standards (National Geographic Resource Science SubCenter, http://gre.geodata.cn. Available P was graded as level 4 with the content of 6.79–24.27 mg kg$^{-1}$. Soil density suitable for plant growth is generally within 1.14 to 1.26 g cm$^{-3}$. In ours research, the average soil bulk density of grasslands in Qilian mountain was 1.01 g cm$^{-3}$, with a value of between 0.75–1.14 g cm$^{-3}$ in most areas of the Qilian Mountains. The grassland soil comprehensive fertility index decreases from east to west. The spatial distribution and succession of grassland types decided the grassland soil fertility. From west to east, the grassland types are desert, typical grassland and meadow grassland mainly. As an indicator of dispersion degree of the sample, CV <10% means weak variation, 10–100% means medium variation and >100% means strong variation. The results of our studies indicated that, except for soil pH, which were weak variations, all the nutrient indicators were medium variable.

Grassland type is determined by climate, vegetation and soil (*Hu, Zhang & Nan, 1978*). As the substrate of grassland, soil physical and chemical properties of different types of grasslands provide important insight to understand grassland evolution (*Gou et al., 2019*; *Li et al., 2019*). *Zhang et al. (2019)* found that the contents of total N, organic carbon and soluble organic carbon of different alpine types of grasslands were in an order of alpine meadow>alpine meadow grassland>alpine grassland>alpine desert, and the differences between various alpine types of grasslands were significant. This study observed that the ranking of the different types of grasslands was desert type >meadow type >steppe type for soil bulk density, the ranking of total porosity were opposite to that of soil bulk density. Furthermore, the total N, SOM and soil comprehensive fertility index in different grassland types had significant differences. Since soil nutrients were mainly derived from the decomposition of animals, plants, microbial residues, litters, root exudates and soil parent materials, spatial heterogeneity of soil fertility distribution in different types of grasslands observed in this study indicated these grasslands were influenced through the different climate and vegetation (*Wei, Zhou & Shi, 2018*). Soil organic matter mainly came from the decomposition of organic residual, but moisture and temperature were the dominant factors controlling the decomposition rate of organic matter. This was why *Ren et al. (2008)* used precipitation and temperature accumulated as a first-class classification index to classify grassland types in Comprehensive and Sequential Classification System (*Ren et al., 2008*).

Evaluation factors affect the rationality and objectivity of evaluation results to a certain extent (*Chen et al., 2019*; *Soil Science, 2019*). In many studies, the evaluation indicators of soil fertility mainly focused on nutrients such as soil organic matter, nitrogen, phosphorus and potassium (*Chen et al., 2019*; *ScienceSoilScience, 2019*; *Yu et al., 2018*). The soil bulk density and total porosity can reflect the status of soil fertility from different angle as soil

compactness, permeability, infiltration performance and water holding capacity (*Garrigues et al. , 2012*). The modified Nemerow formula highlights the effect of the minimum factor on soil fertility, reflecting the law of the limiting factor of plant growth in ecology (*An et al., 2015*), and the soil minimum factor can be identified according to the minimum value of the Fi in Nemerow formula. In ours study, the soil fertility of different types of grasslands had different main limiting factors. Such as pH, total N and SOM were the main factors limiting soil fertility in LM, and pH and available P were the main factors limiting soil fertility in UM, AM, TS and AS. The Soil bulk density, pH, total N, SOM and available P were the main factors limiting soil fertility in TD and TDS. Soil bulk density and available P were the main factors limiting soil fertility in AD. Nemerow index method can objectively reflect the comprehensive fertility characteristics of grassland soil, but many studies have not analyzed the spatial distribution characteristics of soil fertility in depth (*Bao et al., 2012*; *Fan, Li & Wu, 2012*). Ours research combined GIS and soil science to draw a spatial distribution map of grassland soil fertility in Qilian mountain, which more intuitively reflected the distribution of grassland soil fertility. In ours study, the areas with high, medium and low soil fertility accounted for the total grassland area of Qilian Mountains was 45.60%, 41.92% and 12.46%.

Grassland was an important foundation for the construction of the Qilian Mountain ecosystem. Based on the research results, the actual distribution of grassland types, and reasonable management could promote benign and sustainable development of grassland ecosystems.

## CONCLUSIONS

The results of soil fertility indexes and their spatial distribution of the grasslands in Qilian mountain showed that, except for the low-available P content, all the soil fertility indexes had reached level 2 and above according to China's second soil census standard, while soil bulk density was relatively low and pH was relatively high. The soil comprehensive fertility index was in a ranking order of TS >AM >AS >UM >AD >LM >TDS >TD, and the areas with high, medium and low soil fertility accounted for 63.19%, 34.24% and 2.57% of the total grassland area respectively. Soil fertility of different grassland types had different main limiting factors, for instance, the pH, total N and SOM were the main factors limiting soil fertility in LM, while they were pH and available P for UM, AM, TS and AS. The typical grasslands and meadows may need to apply phosphorus, and desert grasslands to apply both nitrogen and phosphorus to improve comprehensive soil fertility and grassland productivity.

### Funding

This study was supported by the National Natural Science Foundation of China (31160475, 61401439); and a new round of grassland reward and subsidy benefit evaluation and grassland ecological evaluation in Gansu Province (XZ20191225). The funders had no role in study design, data collection and analysis, decision to publish, or preparation of the manuscript.

### Grant Disclosures

The following grant information was disclosed by the authors:
National Natural Science Foundation of China: 31160475, 61401439.
Grassland Reward and Subsidy Benefit Evaluation.
Subsidy Benefit Evaluation and Grassland Ecological Evaluation in Gansu Province: XZ20191225.

### Competing Interests

The authors declare there are no competing interests.

### Author Contributions

- Qiang Li conceived and designed the experiments, performed the experiments, analyzed the data, prepared figures and/or tables, authored or reviewed drafts of the paper, and approved the final draft.
- Junyin Yang, Wenhao Guan, Zhigang Liu and Guoxing He performed the experiments, prepared figures and/or tables, and approved the final draft.
- Degang Zhang and Xiaoni Liu conceived and designed the experiments, prepared figures and/or tables, and approved the final draft.

### Data Availability

  Raw data are available as Supplementary Files.

### Supplemental Information

Supplemental information for this article can be found online at http://dx.doi.org/10.7717/peerj.10986#supplemental-information.

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
