# Peer review of "Soil fertility evaluation and spatial distribution of grasslands in Qilian Mountains Nature Reserve of eastern Qinghai-Tibetan Plateau"

_PeerJ, doi:10.7717/peerj.10986_

## Round 0.1 · original submission · Major Revisions

The article contains some information valuable to know the soil fertility status of an area of considerable environmental value and high vulnerability (according to the information provided by the authors). As reviewer 2 states, “the results are relevant not only for the management of the grasslands in the Tibetan Plateau but also for our understanding of grassland ecosystems”. Nevertheless, I agree with the three reviewers that the manuscript needs major revisions before being acceptable for publication.

First of all, the objectives of the study should be more clearly defined and the use of the Nemerov index (developed to assess pollution) to evaluate soil fertility should be justified, providing a bibliography accessible to a non-Chinese-speaking reader. Moreover, the use of variables such as total phosphorus or total potassium as indicators of soil fertility does not seem appropriate.

As two of the reviewers point out, the methods are not sufficiently described, particularly the analytical methods. In various cases, the given reference does not correspond to the measured variable. There are even mistakes, such as saying that available P is determined upon acid digestion. It is not clear how many samples were taken at each sampling site, nor how many samples in total. The authors say in the Materials and Methods section that soil was sampled at three depths, but no reference to sampling depth is made in the Results or Discussion.

The calculation of the Nemerov index must be explained in detail. Partial indices are calculated, corresponding to each of the assessed variables. Apparently, the value of each partial index is higher the higher the value of the measured parameter. However, the parameters that determine soil quality (one of whose components is soil fertility) can be of the type "more is better", "less is better" or "optimal range". Although it is not clear in the discussion, it seems, for example, that the calculated fertility index is lower for soils with lower bulk density, which does not seem logical.

The classification of soils, particularly soils corresponding to each grassland type, according to a widespread classification system (WRB, Soil Taxonomy) is lacking.

The authors refer to degradation of grasslands, but there are no results for the fertility index calculated for soils in different degradation degrees.

There seems to be some confusion between soil fertility and soil quality (anyway, total phosphorus and total potassium are not indicators of either concept).

The English language needs to be improved. Care must be taken with the punctuation marks.

Some references in the text are not in the list of references and vice-versa. Some references in the list of references are duplicated. Other references are incomplete or contain errors.

Detailed comments are included in my annotated manuscript.

I fully agree with all the comments by the three reviewers. Please take them into account.

·

Basic reporting

1. Basic Reporting

The English used for writing this manuscript requires improvement. The authors conducted out this research in a scientifically accepted way; however, it lacks international interest. The finding reported are mostly of local interest. There are some differences with the commonly accepted definitions used in the western world. For instance, the concept of Soil Fertility that is included in the title does not matches with the concept studied in this paper..
.

The manuscript presents a list of appropriate references titled. However, most of then are in Chinese, and this reviewer could not confront them.
Professional article structure, figures, tables. Raw data shared.
The structure of the article conforms to a commonly acceptable format for reporting this type of research.

Figures are relevant to the content of the article; however, the resolution is not sufficient for publishing. Figures titles are adequate.

The manuscript does not formally present the description of the hypotheses. They are self-contained in the objectives.

Experimental design

The research fulfills the scope of the journal. However, as previously indicated, its interest is located mainly in the area where the authors experimented.
Research question definition, relevance, and importance.

The research questions are relevant and meaningful for the local environment. It is explicit and legitimate, trying to understand the reason how the prairie soil fertility changes with position in the landscapes and affects its development and performance. However, its interest is limited to the experimental zone and can hardly be extrapolated. The most interesting aspect, that is, the use of the Nemerow method, no commonly used for soil fertility studies, it is not well explained.

The experimental procedures described are usually the ones employed in this type of research. There is not an indication of unethical practices if conducted as described.

The chemical methods applied for conducting the soil analysis are not described at all (like avaialable-P) or the described techniques are not those employed to study soil fertility. It appears the authors are using the total content of N, P, and K as soil fertility indicators incorrectly. The entire content of these nutrients is soil fertility indicators of the potential amount, but they are not of the indexes of the available source for plant growth.

This reviewer considers that the sampling procedure was not the appropriate one. The spatial soil variability requires a more comprehensive sampling method. If we take into consideration the size of the experimental units, more samples must be taken to have a complete picture of the spatial variability of the soil.

Validity of the findings

The impact and novelty of the reported findings are described, however the authors thinks that all potential lectors are familiarized with some of the Chinese developments. These aspects has to be clearly explained. The authors require to improve the discussion of some of their results. In addition, The English .must be improved.

The statistic used is simple but appropriate. A different planning of sampling activities could have improved the scope of the work.

There is not a good agreement between the objectives planned and the conclusions. See comments on the attached letters to the Editors and the authors with commentaries.

Additional comments

Review
Soil fertility evaluation and spatial distribution characteristics of grassland in Qilian mountain nature reserve
Qiang Li Corresp., 1 , Junying Yang 1 , Wenhao Guan 1 , Zhigang Liu 1 , guoxing he 1 , Degang Zhang 1 , Xiaoni Liu 1


Comment 0 Line 1. Soil fertility...... Please review the title. The parameters described are not commonly used to evaluate soil fertility. Total N, P, and k are parameters used to describe the chemical characteristics of the soil
Comment 1. Line 3. Check the way you wrote the name of one of the authors (guoxing he 1).
Comment 2: Line 10. Total N, total K, total P, and available K were graded at level 2 (high) or above with content of 4.99-131.52, 0.63-4.97, 15.36-21.72, 0.81-1.69 and 0.21-1.06 g kg-1 respectively for most regions. ………… For assessing actual soil fertility, the total content of any essential elements for plant growth prefers the use of the available fractions. The purpose of determining the full content of a soil chemical component is mainly for chemical characterization. They are not directly related to nutrient availability for plants.
Comment 3. Line 14. The soil comprehensive fertility index was in a ranking order of Temperate steppe >Alpine meadow > Alpine steppe >Upland meadow >Alpine desert>Lowland meadow>Temperate Desert Steppe >Temperate Desert and the areas with high, medium and low soil fertility accounted for 63.19%, 34.24% and 2.57% of the total grassland area of Qilian mountain soil fertility for different grassland types had different main limiting factors. ………… Hard to read sentence. Plese rewrite or split the sentence into two and remove unnecessary words. Use the active voice if possible.
Comment 4. Line 17. The grassland soil fertility of Qilian mountain was generally at the mid-upper level ……. The chemical tests performed are not the best indicators of soil fertility. Therefore, it is not easy to draw this conclusion.
Comment 5. Line 18. Considering the main limiting factors of different type grasslands and spatial distribution, the typical pastures and meadows may need to apply acidic phosphate fertilizers, and desert grasslands to apply compound fertilizers of nitrogen and phosphorus to improve soil comprehensive fertility to improve grassland productivity………… Please consider phrasing this sentence in the following way: The main limiting factors found in the different types of grasslands and spatial distribution, the typical pastures and meadows may need to apply acidic phosphate fertilizers, and desert grasslands to apply compound fertilizers of nitrogen and phosphorus to improve comprehensive soil fertility to improve grassland productivity
Comment 6. Line 33. Soil organic matter, nitrogen, phosphorus, potassium, soil density, and pH are important components of soil fertility, while their content and spatial distribution directly affect grassland vegetation productivity (Wuest, 2015; Li et al., 2014)………. This reviewer only partially agrees with this sentence. Even though N, P, and K are related to soil fertility, are their available forms which the authors must consider as directly related to an active concept of soil fertility.
Comment 7. Line 33. ……nitrogen, phosphorus, potassium…. available nitrogen, phosphorus, potassium….total content are part of the capacity component of the system.
Comment 8. Line 33….. soil density and pH are important components of soil fertility, while their content… Please, revise
Comment 9. Line 331-334. Hu S J, Hu R, Pu Y L, et al. 2018. Influence of ecological restoration on soil biological fertility in desertified 332 grassland[J]. Pratacultural Science. 333 Hua S, Li X, Jinting Z, et al. 2018. Evaluation of Heavy Metal Pollution in the Soils around the Brownfield Based 334 on the Modified Nemero Index Method[J]. Environmental Protection Science. ….. Incomplete citations.
Comment 10. Line 291-292. Bao Y X, Xu M G, Li F T, et al. 2012. Evaluation Method on Soil Fertility Under Long-Term Fertilization[J]. scientia agricultura sinica,. Incomplete citation and erroneous form to cite the authors.
Comments 11 Line 398- 300. Wang G H, Ren J Z, Zhang Z H. 2001. A Study on the population diversity of plant community, in Hexi mountain399 oasis-desert area: General features. ACT A PRATACULTURAE SINICA, 10(1): 1-12….. You must be consistent in the way of citing.
Comment 12 Line 38 …… index method has been well recognized due to it it can avoid the influence…. Check your writing.
Comment 13 Line 289 An K, Xie X P, Zhang H Z, et al. 2015. Spatial pattern and impact factors of soil fertility in West Lake Scenic Area. 290 Chinese Journal of Ecology, 34(4): 1091-1096. (in Chinese) DOI: 10.13292/j.1000 -4890.20150304.020 ….. Too many articles in Chinese. Cannot be checked properly.
Comment 15 Line 47-48. Grassland ecosystem is the largest ecological system in Qilian mountain natural reserve. ….. How large?
Comment 14 Line 52. There are many types of grasslands in Qilian mountain… this sentence does not give much information. Be specific. which types of grasslands
Comment 15 Line 58. …….the aims of this study were to investigate the soil of different type ??? grasslands in Qilian mountain natural reserve, 2) to analyze the distribution characteristics of soil fertility index ????, and 3) to determine the limiting factors for grassland soil fertility in Qilian mountain nature reserve, which can help to provide scientific insight for improving grassland ecological services.
Comment 16 Line 272. Conclusions. The results of soil nutrients???? and their spatial distribution of grasslands in Qilian mountain showed that, except for the low-available P content, other nutrient indicators ????? had reached level 2, and above in China's second soil census standard, while soil bulk density was relative low and pH was relatively high (soil nutrient???). The soil comprehensive fertility index was in a ranking order of TS > AM > AS > UM > AD >LM > TDS > TD (Do not use abbreviations in the Conclusions), and the areas with high, medium and low soil fertility accounted for 63.19%, 34.24% and 2.57% of the total grassland area of Qilian mountain. The different type grasslands soil fertility had different main limiting factors ?????? The grassland soil fertility of Qilian mountain was generally at the mid-upper level ??????. The appropriate management methods ????? should be adopted to improve soil fertility combining with the main limiting factors, to improve grassland ecological service function. …….These conclusions are not informative and are not quite in line with the objectives or aims proposed.
Comment 17 Line 88 .. 1.3 Sample analyses………………. Poor descriptions of the chemical methods. If anybody would like to replicate this experiment could hardly replicates the same procedures. Good description of the experimental condictions is the base of the good science.
Comment 18 Line 66 ….in the order of forest, shrub, grassland and desert. …… in the following order forest, shrub, grassland and desert.
Comment 19 Line 67… At vertical direction, from low to high altitude…….. Provide the altitudes.
Comment 20 Line 68 The main types of soil are mountain gray cinnamon soil, subalpine meadow soil, alpine meadow soil and alpine cold desert soil. …….. Use WRB o Soil Taxonomy and the equivalent soils.
Comment 21 Line 77 The sampling time….. The soil sampling time???
Comment 22 Line 77 The central points……The central area…….???
Comment 22 Line 80. ..…. each sample spot using soil drill at a…….. each sample point using soil drill at a…. Do you mean an auger or an electric drill??
Comment 23 Line 81 at a depth of 0-10 cm, 10-20……..do you mean at a depth increments of 0-10 cm, 10-20
Comment 25 Line 85 The four soil samples from each layer were mixed as one sample for that layer…. The four soil samples from each depth increment were mixed to compose one sample.
Comment 26 Line 79-84 Please rewrite your sampling desing. It is not clear.
Comment 27 Line 94 Total P was measured by Mo-Sb colorimetry ……… Error. The indicated colorimetric method measures the color of any form of P in solution. This method does not measure total P. For measuring total P you mast first destroy the soil matrix to place the P in solution. The method you mention is for developing a color that is associated, according with the intensity of the blue color developed, with the P concentration in a solution.
Comment 28 Line 96-98 Available P was determined by the Mo-Sb-Ascorbic acid colorimetric method after acid digestion (Nelson and 97 Sommers, 1996)………. Your concept of availability es erroneous. You are not indicating a method to measure available P and you are reporting available P data ?????
Comment 30 Line 106 Table 2. Classification criteria used for soil nutrients ….. Soil C is not a nutrient. You did not mention any method for determining organic carbon. What you measured was readily oxidizing organic matter and estimated the organic C from these data.
Comment 31 Line 116 The improved Nemerow index method was used to……. The authors do not provide the original paper with the method description. Impossible to check.. Instead of information on users of the methods is presented.
Comment 32 116 Line The improved Nemerow index method was then used to evaluate the grassland comprehensively….. Does the previous sentence imply that other interpretation methods of soil fertility are not comprehensive? The author must explain why they argued that the proposed way is better than the traditional ones.
Comment 33 Line 41. This method is more accurate than many other interpolating methods and can also effectively avoids systematic errors……….. The authors should explain why this method is more accurate.
Comment 34 Line 144 There is note here… (Provide a reference here).
Comment 35. Line 160 The total porosity was 35.36-58.83%... Provide significant figures only here and throughout the text.
Comment 36 Line xxx The available P was 6.79-24.27 mg kg-1, with an average value of 12.81 mg kg-1 and a CV of 43.09%. The available K was 0.21-1.06 g kg-1, with an average value of 0.40 g kg-1 and a CV of 68.00%...... Information on how did you measured these parameters must be provided. The figures reported do not have any meaning if the method of determination is unknown.
Comment 37 Line 172 The soil fertility index for different type grasslands were shown in Table 5……. are shown…..
Comment 38 Line Consider presenting the information given in this paragraph differently. The present way is tedious and does not help to the understanding of the paper.
Comment 38 Line 189 2.3 Soil physical and chemical spatial distribution….. the maps are a nice way of presenting the results.
Comment 38 Line 202 The soil comprehensive fertility index…… the meaning of the soil comprehensive fertility index should be clearly explained.
Comment 38 Line 234. Soil organic matter is closely related to soil fertility and soil health. Nitrogen, phosphorus, and potassium provide essential nutrients for plant growth and development, and are the main components of soil nutrients…….. Soil organic matter content is closely related to soil fertility …… The available forms of nitrogen, phosphorus, and potassium provide essential nutrients…….
Comment 38 Line 236 The content of soil organic matter, total N, total P and available K were graded as level 2 (high) or above for Qilian mountain grasslands according to the classification of China second soil census standards… If the authors do not explain clearly the meaning of the Chinese classification, this information has scarce meaning. Is an arbitrary classification or is based on some previous correlation studies with planta productivity?
Comment 38 Line 236 The content of soil organic matter, total N, total P and available K were graded as level 2 (high) or above for Qilian mountain grasslands according to the classification of China second soil census standards, with content of soil organic matter, total N, total P and available K of 4.99-131.52, 0.63-4.97, 0.81-1.69 and 0.21-1.06 g kg-1 for most regions.Available P was graded as level 4 with the contents of 6.79-24.27 mg kg-1. ,,,,,,,,, This sentence is hard to read.
Comment 38 Line 240 Ours study found that the…….. Our study……. or our studies
Comment 38 Line 242 …..which means there was a background difference in the distribution of soil nutrients ….. there was a background difference does not have a clear meaning.
Comment 38 Line 245 The average soil bulk density of grassland in Qilian mountain was 1.01 g cm-3, with most of the area between 0.75-1.14 g cm-3. The grassland soil comprehensive fertility index of Qilian mountain decreases ……. This asseveration requires an interpretation. Think on the role of root density in the grassland.
Comment 38 Line 251 The results of ours study indicated… …. The results of our studies indicated…….
Comment 38 Line 253-294. Improve English. There too many passive voices. The passive voice is no incorrect, but most Editors prefer not to use this voice in scientific writing.
Comment 38 Line 255 More important than ranking the different types of vegetation is explaining why occurred the differences.
Comment 38 Line 251 The rational construction of comprehensive evaluation factors was the key content of comprehensive evaluation of soil fertility, which directly determined the rationality and objectivity of evaluation results ….. Wordy sentence. Please review it.
Comment 39 Line 251 See Commrnt 16.

·

Basic reporting

The article reports the results of a good study, but it needs improvement in many parts and would greatly benefit from English editing. In particular, the introduction, methodology, and results and discussion need to be revised and supported with more scientific literature.

Experimental design

The research question is relevant. Moreover, the design and methodology of the study are valid. However, the methodology needs to be added with more details. Specifically, the steps of the soil fertility evaluation need to be clearly stated. Also, more details should be given on the relatively less known Nemerow index which basically assigns ratings to selected soil properties. The study appears to have been carried out well.

Validity of the findings

The data appear to be of good quality although the interpretation should be improved. Likewise, the conclusions need to be revised to answer the objectives of the study. The results are relevant not only for the management of the grasslands in the Tibetan Plateau but also for our understanding of grassland ecosystems.

Additional comments

Comments:
Line 1: The title may be revised to “soil fertility evaluation and mapping of the grasslands in the Qilian mountain nature reserve in the Tibetan Plateau”.
Lines 6-23: The Abstract needs to be revised after the manuscript has been improved.
Line 30: This needs improvement. The authors may cite literature on the topic such as the paper by Wardle et al. 2004. Science 304: 1629-1633.
Line 32: Delete the word “reconstruction” which appears not appropriate
Line 35: Change soil density to soil bulk density
Lines 37-40: The widely recognized methods of soil fertility evaluation are soil test, plant tissue test, nutrient deficiency symptoms, etc. Please see Foth 1990. Fundamentals of Soil Science (8th edition).; Marschner. 1995. Mineral Nutrition of Higher Plants (2nd edition).; Jones 2013. Plant Nutrition and Soil Fertility Manual (2nd edition).
Lines 40-44: The Nemerow index is an indicator of soil fertility based on the soil test method. Correlation coefficient and principal component analysis are not methods of soil fertility evaluation. Since the Nemerow index is not widely known and the reference cited Bao et al. (2012) is not easily available, it is important for this index method to be described in terms of its origin, principle, applications.
Lines 47-50: There have been a lot of publications on the use of GIS for soil evaluation. Aside from the Chinese studies cited, it would improve the quality of the manuscript if other international publications are cited. The authors may read Brevik et al. 2016. Geoderma 264; Liengsakul et al. 1993. Geoderma 60; Miller 2017. In: Soil Mapping and Process Modeling for Sustainable Land Use Management.
Lines 72-80: There is a need to include information on geology, soil classification (USDA or WRB), climate classification according to Koeppen.
Line 87: Is this grassland classification from China or is it an international one? Please cite references.
Lines 88-89: Change “sample line” to transect; change “sample site” to sampling sites; change “sample spots” to sampling points.
Lines 96-97: In Table 1, some scientific names have no authors; some authors of scientific names are inconsistently written (example: L. or Linn.). The names of authors of scientific names should not be italicized.
Line 111: Evaluation of soil fertility. There is a need to describe clearly the steps of the evaluation method used in the study. Please explain how the individual indicators are related to/used in the comprehensive evaluation. Please describe clearly the steps of the Nemerow index method so that the readers will be able to follow it.
Line 112: Is it soil organic matter (SOM) or soil organic carbon (SOC) as shown in Table 2?
Line 118: In the last column of Table 2, the “comment” should be changed to “interpretation”
Line 121: What is the “improved” Nemerow index method?
Line 124: “to eliminate the dimensional differences between the parameters” is not clear. What does it mean?
Lines 150-151: What is the basis or reference for these classes? (low, medium, high)
Lines 162-170: There is no need to mention all the values in Table 4. Just focus on the highlights or the most important results.
Lines 175-184: It would be more meaningful if it is stated clearly what grassland type has the lowest values of bulk density, etc., which have the highest values. Using all the symbols such as AM, UM, LM, etc. is not easy for the readers to understand.
Line 187: In Table 5, the note at the bottom should be in the same “column” not row.
Line 223: To identify the limiting factor, what should be the value of Fi in Table 6? Is it below 2.0? Is it below 1.5? Or is it just the lowest in a particular grassland type?
Lines 236-297: In the Discussion, there is a need to explain the choice of the soil properties selected. Why are total P and total K included although these two parameters are weakly correlated to their availability to plants? Please cite references to support discussion. There is also a need to explain all the trends found. For example, there is a need to discuss why some grasslands have high or low SOM values, low or high pH values, etc. Please support discussion with citations, if possible from international publications (in addition to the Chinese publications already cited).
Lines 298-307: Conclusions should be revised after the whole manuscript has been improved. The conclusions should answer the objectives of the study.

·

Basic reporting

I am not able to assess the English used in this paper
Literature references are sufficient
The introduction is very scarce
The structure is correct
Figures are not sufficient

Experimental design

The research fits within the objectives and scope of the journal
The methods are not described in sufficient detail

Validity of the findings

No comment

Additional comments

In my opinion, it is necessary to improve the writing of the paper and on the other hand, it is necessary to incorporate, if possible, climate data to reach more precise conclusions. Before publication, the paper needs profound transformation.

---

## Round 0.2 · Major Revisions

The manuscript has somehow been improved with respect to the previous version. Nevertheless, it still needs important revision before been published.

The English language still needs to be improved (it is not better than in the previous version).

The rebuttal letter is not very informative. To most of the reviewers’ comments, the authors just answer “we had amended”. To the recommendation of one of the reviewers for thinking of the role of root density, the authors surprisingly answer “we would be thinking on the role of root density in the grassland ¡in future!”. Or to one recommendation on the language, they say “We will improve language expression further”. More surprisingly, to other reviewer's recommendation, the authors answer “Maybe your suggestion is good, we have to consider”. There are no answers to the editor’s comments.

You have not heeded the recommendation to use a widespread soil classification system.

Available K and P are determined using HF-HNO3-HCl04. This is quite surprising.

The text refers to Table 6, but Table 6 does not exist.

You have removed total K and total P as indicators, but the percentages of areas with high, medium and low soil fertility are exactly the same as in the previous version (63.19%, 34.24% and 2.57%). How can you explain this?

I agree with reviewer 2 that the manuscript still needs major revisions. Please refer to the comments by the reviewers, to my comments to the previous version and to my comments in the new annotated manuscript. Please respond precisely to each comment in the rebuttal letter.

·

Basic reporting

The authors considered most of the suggested modifications.. I marked in colours a few items that must be checked again. In addition, the Summary was reviewed and a better English was introduced.
This reviewer suggest to authors to give the manuscript to a native English professional translator for improving the written language.
Check some of the new references introduced. i found at least one mistake.

Experimental design

It is OK

Validity of the findings

It is OK

Additional comments

The authors need to improve the English.
See colored marks in text.
Check the new literature introduced in the manuscript

·

Basic reporting

As in my previous review, the article reports the results of a good study but it needs improvement in many parts and would require English editing.

Experimental design

The methodology is scientifically valid but requires a major improvement as I clearly indicated in the comments. Unfortunately, most of my comments were ignored during the revision of the manuscript.

Validity of the findings

The data are valid.

Additional comments

Title: SOIL FERTILITY EVALUATION AND SPATIAL DISTRIBUTION OF GRASSLAND IN QILIAN MOUNTAIN NATURE RESERVE-TIBETAN PLATEAU

I have carefully reviewed the revised manuscript, but unfortunately, most of my comments and suggestions to improve the quality of the manuscript were either ignored or not satisfactorily addressed by the authors. These are as follows (line numbering is based on the original manuscript):

Lines 40-44: The Nemerow index is an indicator of soil fertility based on the soil test method. Correlation coefficient and principal component analysis are not methods of soil fertility evaluation. Since the Nemerow index is not widely known and the reference cited Bao et al. (2012) is not easily available, it is important for this index method to be described in terms of its origin, principle, applications.

Lines 72-80: There is a need to include information on geology, soil classification (USDA Soil Taxnomy or World Reference Base system), climate classification according to Koeppen.
(The soil names like cinnamon soil, meadow soil, etc are not acceptable soil names).

Line 87: Is this grassland classification from China or is it an international one? Please cite references.

Line 111: Evaluation of soil fertility. There is a need to describe clearly the steps of the evaluation method used in the study. Please explain how the individual indicators are related to/used in the comprehensive evaluation. Please describe clearly the steps of the Nemerow index method so that the readers will be able to follow it.

Line 121: What is the “improved” Nemerow index method?

Line 124: “to eliminate the dimensional differences between the parameters” is not clear. What does it mean?

Lines 150-151: What is the basis or reference for these classes? (low, medium, high)

Lines 162-170: There is no need to mention all the values in Table 4. Just focus on the highlights or the most important results. (If you mention all the values in the text, then the Table is not necessary. It is redundant and not good scientific writing).

ADDITIONAL COMMENTS based on the revised manuscript:

Lines 83-85: It is not clear if clod samples or core samples were collected for bulk density determination. What is “ring knife method”? How is it different from the core method?

Line 89: “small scale” Kjeldahl should be changed to micro Kjeldahl

Lines 92-94: HF-HNO3-HCl04 is not for available P or available K

Lines 151-154: It says that the bulk density values ranged from 0.75 to 0.94 g/cm3 and the total porosity of 50 to 60%. If you calculate total porosity from the bulk density values obtained and the standard particle density of 2.65g/cm3, the total porosity would range from 64 to 72%. The point is this: bulk density values below 1.0 g/cm3 indicate a very high soil porosity.

Finally, this reviewer humbly suggests that the manuscript be edited by an English editor, preferably a native English speaker before it is re-submitted.

Recommendation:
This reviewer recommends that the manuscript be returned to the authors for major revision.

·

Basic reporting

The paper has certainly improved. Possibly English need to be revised
I recommend the publication of this paper.

Experimental design

The description of the methodology has also improved.

Validity of the findings

The conclusions are well established and linked to the results.

---

## Round 0.3 · Minor Revisions

I agree with the reviewer 2 that the article has improved by still needs minor revisions. Besides the comments of reviewer 2, with which I fully agree, I enclose some remarks in the annotated manuscript; most of them were already in the previous reviewed version.

·

Basic reporting

Please see previous comments

Experimental design

Please see previous comments

Validity of the findings

Please see previous comments

Additional comments

Comments

This second revised version of the manuscript is very much improved and incorporates most of the reviewers’ comments and suggestions. I commend the authors for exerting much effort in improving the manuscript. However, the manuscript still needs minor improvement in its technical content (please see specific comments below). Moreover, the manuscript needs further English editing preferably by a professional English editor to make it more understandable to Peer J’s global readers. In particular, there are several vague and inconsistent statements in the Methodology and Discussion portions that need to be corrected.

Specific comments:

Line 12: change “soil comprehensive fertility index” to comprehensive soil fertility index.
Line 20: change “may demand acidic phosphate fertilizers” to may require phosphate fertilizer application
Line 23: add in the keywords Nemerow Index and soil fertility index
Line 47: give the full meaning of AHP
Line 54: this statement needs to be improved. It is difficult to understand
Lines 55: “ law of limiting factor”. Are you referring to Liebig’s Law of the Minimum?
Line 99: please indicate the references/authors of the methods used for available P and K.
Line 121: “soil worst attribute”. Do you mean the most limiting soil fertility attribute?
Lines 120-125: this statement needs improvement to make it clearer.
Lines 197 and 202: change “decided” to “determined”.
Line 213: “residual organic matter”. Do you mean organic residue?
Line 222-223:” law of the smallest factor”. Do you mean Liebig’s Law of the Minimum?
Lines 243-245: there is a need to improve the conclusion. Specifically, it needs to indicate the limiting soil fertility factors (please see the Abstract where it is much better stated).

---

## Round 0.4 · accepted · Accept

The reviewers and I did our best to help you improving the manuscript. I think you could have gotten more out of our advice.